# Reproducibility Report of LaneATT

**Guide Ai**         **Yifeng Cao**         **Jiaming Zhong**

## Reproducibility Summary

This report is meant to reproduce the result based on the paper: *Keep your Eyes on the Lane: Real-time Attention-guided Lane Detection*, where the authors built an efficient and accurate lane detection model - LaneATT.

**Scope of Reproducibility**

The original paper claimed that among all state-of-the-art methods, LaneATT achieves high accuracy while maintaining real-time efficiency. The central parts of the methodology was also tested and it shows that while increasing accuracy, the novel anchor-based pooling method allows lighter backbones. While there is no visible markings the attention mechanism it proposed increase the performance significantly. Comparing to Cross-Entropy Loss, Focal Loss is computation efficient and shows beneficial for accuracy. Each claim was tested with different experiments.

**Methodology**

The model was tested on two datasets: TuSimple and CULane. Apart from the model's hyperparameters, the image size and the number of anchors also influence the result. In prepossesing, all the images from the two datasets was resized to same scale and the annotations were also translated into anchor based points.

Seven experiments was designed to evaluate the claims in the scope of reproducibility. The experiment based on the paper's configuration was first tested. Then the efficiency of anchor representation, anchor-based feature pooling, the attention mechanism, the loss function was evaluated. Lastly the application of other backbone was also tested.

**Results**

The result shows that with the experiment based on the paper's configuration, the same result can be obtained. Further experiments also shows that the anchor-based pooling method can significantly increase accuracy. While applying the attention mechanism, the speed becomes slower but there is only a tiny increase in accuracy. The comparison of different loss function shows that the Focal Loss achieve only slight improvement comparing to Cross-Entropy Loss. Cross-Entropy Loss can also work perfectly. Comparison of different backbones shows that the model enables the use of light weight backbones.

**What was easy**

Based on the following datasets: TuSimple and Culane, the experiments based on the author's original code can be easily setup and the result can be nearly the same as the paper claimed.

**What was difficult**

The annotations for the testing dataset of LLAMAS can not be obtained, therefore the experiment on this dataset can not be done.

When testing the effectiveness of the anchor-based feature pooling method, the ablation experiment of the proposed method is the same as the model of Line-CNN, while the code of Line-CNN is not open sourced, modifying of the LaneATT code might be needed.

**Communication with original authors**

None

33rd Conference on Neural Information Processing Systems (NeurIPS 2020), Vancouver, Canada.

# 1   Introduction

The development of autonomous driving requires the autonomous vehicles to be safer for both passengers and pedestrians, such request constrained the vehicle to equip with better perception systems, in further detailed, a better lane detection system which allows the vehicle to knowledge the exact position of the lanes. A good lane detection model need to resolve many challenging problems, as in real-world scenario, it needs to be robust under different lighting, weather and lane mark conditions. More importantly, the lane detection model has to be a real-time application, meaning it should consume less computing power while provide reliable results within acceptable time [1].

To address the stated problem, a variety of researches have been conducted. Different from tasks detecting other objects, lanes have unique geometric characteristics for their long thin structures. Early works mainly relies on handcrafted filters for line features extraction. Recent works relies mostly on deep neural networks for detecting and predicting lanes. The selected paper proposed a one-stage region-based detection method, . It allows the use of lightweight backbones by applying a anchor-based pooling method that can effectively extract feature from the output of the backbone. Thus the method proposed by the paper can achieve high accuracy and also real-time efficiency.

# 2   Scope of reproducibility

The authors of the paper: *Keep your Eyes on the Lane: Real-time Attention-guided Lane Detection* proposed an anchor-based single-stage lane detection method -LaneATT. It was built based on Line-CNN [2] where the feature on the three edges of the image plane (left, right, bottom) was extracted by a CNN to verify with traffic lanes that are represented by origin points and direction angles. LaneATT named such representation as an anchor and proposed an anchor-based feature pooling and attention mechanism to increase efficacy and efficiency. The LaneATT was proposed with four central claims.

- Claim 1 LaneATT has the best trade-off between the efficacy and efficiency compared with all state-of-the-art methods
- Claim 2 Anchor and Anchor-based pooling enabled lighter backbones while increasing the accuracy significantly
- Claim 3 The attention mechanism can significantly increases the model performance of lane detection when there is occlusion or no visible lane markings
- Claim 4 Use Focal Loss instead of Cross-Entropy Loss is beneficial for accuracy and computation efficiency

Each claim will be verified in Section 4 below.

# 3   Methodology

The reproduction of LaneATT was done with the open-source code provided by the authors. The ablation experiment of anchor-based feature pooling is the same as the model of Line-CNN. However, Line-CNN does not provide open-source code, it was accomplished by modifying the LaneATT code instead.

## 3.1   Model descriptions

The LaneATT model was built in two parts: a lightweight backbone CNN to achieve high accuracy and a novel anchor-based attention mechanism to aggregate global information. Three different backbones were used: ResNet-18, 34 and 122 to test for the efficiency trade off. ResNet are convolution neural networks and the number indicate the number of trainable layers. The batch size was set to 8, learning rate - 0.0003 with a scheduler as CosineAnnealingLR and the optimizer used was Adam. Each backbone model will run by 15 epochs.

## 3.2   Datasets

Three datasets were used to test the LaneATT: Tusimple, CULane and LLAMAS. The Tusimple dataset contains 6,408 1280x720 pixels highway scenario images and was divided into 3,626 images training set, 258 images validation set and 2,782 images testing set [3]. The CULane dataset contains 133,235 1640x590 pixels city scenario images that contain nine different categories: Normal, Crowded, Dazzle, Shadow, No line, Arrow, Curve, Cross and Night. The dataset was divided into 88,880 images training set, 9,675 images validation set and 34,680 images testing set [4]. The LLAMAS dataset is a large lane detection dataset that contains over 100,000 1276x717 pixels highway scenario

images and was divided into 58,269 images training set, 20,844 images validation set and 20,929 images testing set [5]. However, the annotations for the testing set were not yet published, therefore the result for this dataset is ignored in this reproducibility report. All the images were resized to 360x640 pixels during preprocessing and all the annotations were translated into the anchor based points to be compared with the results. The download link of the datasets can be found in the *DATASETS.md* file.

### 3.3 Hyperparameters

Besides the hyperparameters for training, the proposed LaneATT introduced two more factors that influence the result: the image input size and the number of anchors. The authors experimented with three different image sizes and concluded that the image size would increase the F1 score but drop the efficiency. Furthermore, the number of anchors was tested from 250 to 1250 with a 250 step incrementation. The result indicated that the efficiency decreased with the increase of anchor numbers, however, the F1 score reaches a maximum at 1000 anchors [1]. Since hyperparameter tuning is not part of the scope and will not validate the claims, all hyperparameters were kept by default from the original code.

As mentioned in 3.5, our GPU memory sizes are not enough for Resnet-122. So we have to reduce the batch size. And we also have to reduce the number of epochs due to the time limitation. However, it will not affect the result of reproduction.

### 3.4 Experimental setup and code

Seven experiments were designed to evaluate the claims. The first and second experiments test the models from the paper and train the models based on the paper's configuration.

The third experiment is to find out the effectiveness of anchor-based feature representation. The anchor-based pooling and anchor feature projection will be removed from the original code. The feature map will pass directly to the fully connected layers to determine the potential lane represented by rays. TuSimple dataset will be used and models will be trained by 20 epochs. The result will be evaluated by accuracy (the total correct predicted lane points over the total lane points).

The fourth experiment is to determine the effectiveness of the anchor-based feature pooling. We implement the Line-CNN model just as the paper's ablation study, since the most significant difference between these two methods is the feature pooling method. We only use the feature at each anchor's start point as the whole anchor's feature. Anchors are all pre-defined, so the start points are pre-defined on the feature map's left, right, and bottom borders. We have to change the whole model's matrix dimensions to achieve the algorithm.

The fifth experiment is to find out the effectiveness of the attention mechanism. The attention mechanism is realized by a square matrix connecting anchors. The diagonal of the matrix is 0, and the remaining elements are mutual weights. All the weights in this matrix should be trained in the training process. To test the effect without the attention mechanism, we can set all the weight in the matrix to zero so that there would be no global information for each anchor.

The sixth experiment will compare the focal loss and the cross-entropy loss. The focal loss is built based on cross-entropy loss to address the imbalanced class problem. The cross-entropy loss is calculated by the log of how much the predicted probability diverges from the actual label as shown in equation (1).

$$CE(p_t) = -\alpha log(p_t) \tag{1}$$

The parameter $\alpha$ is the weight assigned to the loss and $p_t$ is the probability that the example matches the actual label. The difference between the cross-entropy loss and focal loss is that focal loss introduced a new term $(1 - p_t)^\gamma$ where $\gamma$ is the hyperparameter to balance the loss between well-classified examples with the incorrect examples. The increase of $\gamma$ will result in the loss being more toleranced when defining the threshold of well-classified examples. TuSimple dataset will be used in this experiment and the result will be evaluated by accuracy (the total correct predicted lane points over the total lane points).

The last experiment is to discover the potential of changing the backbone to other pretrained models. The backbone could be pretrained version. The dimension of the feature map of different backbones may be different, so it may need to be changed accordingly. The reason why we choose VGG-16 is not only because it is very light, and the dimension of the feature map it outputs is the same as ResNet-18/34. So we only need to change the configuration file.

The codes can be downloaded at *https://github.com/MibclAric/LaneATT-reproductability*

### 3.5 Computational requirements

The experiments were run using mainly two sets of hardware: Intel i7-9700/RTX 2070s and Intel i7 10700/RTX 3080. For the ablation experiments that require comparisons on efficiency, the processing speed can be found in the tables in Section 4 below. For the training of ResNet 122 based models, the GPU memory size need to be larger than 10 GB to follow the original code's configurations.

## 4  Results

Based on the four claims from Section 2, and the six experiments designed from Section 4, the result can be found below that will verify the correctness of the claims.

### 4.1 Results reproducing original paper

The testing and training results from reproducing the original paper can be found below.

#### 4.1.1  Result 1: Testing the models from the paper

We firstly testing the models provided by paper's authors, with the results is shown in Figure 1-left. For dataset tusimple and culane, the scores we got are all almost same as paper's results. It proves the Claim 1: LaneATT has the best trade-off between the efficacy and efficiency compared with all state-of-the-artmethods.

#### 4.1.2  Result 2: Training the models use the configurations provided by the paper

Further more, to prove the Claim 1 more firmly, we trained models by ourselves using the same backbones. The results are shown in Figure 1-right. As we can see, The scores are also same as paper's results, except the resnet 122 model, which is beyond out GPU's memory capacity. However, trained models still got high accuracy and only got a little bit less than expectation, even we use very small batch size and epochs to run. So overall, the Lane-ATT got really high scores on these datasets and with these backbones compare with other state-of-the-art methods, which firmly proves the Claim 1.

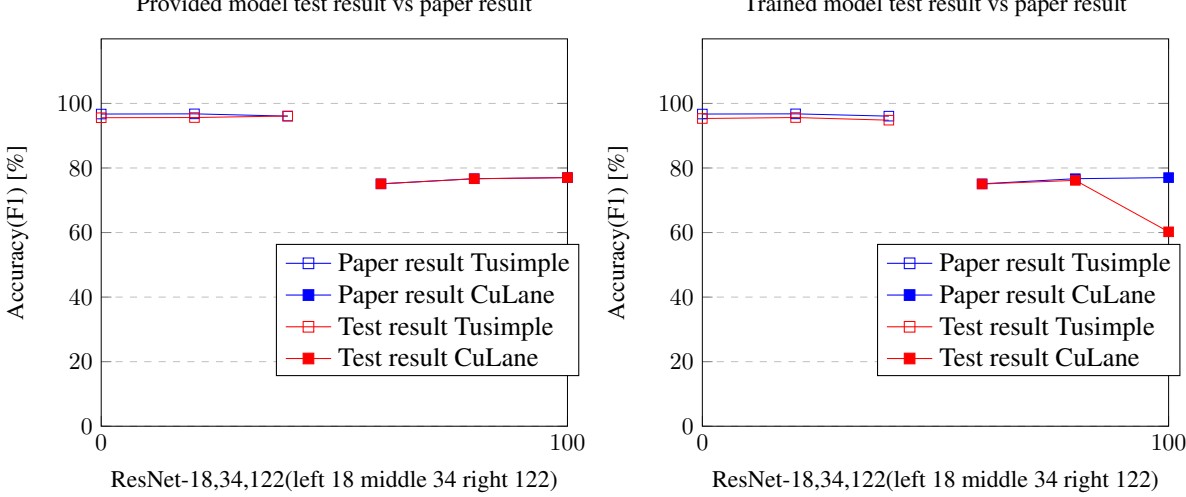

Figure 1: Train and Test result comparison

### 4.2 Results beyond original paper

The five ablation experiment results and the result by using a different backbone can be found below.

### 4.2.1 Additional Result 1: Without Anchor

The result is shown below in Figure 2. It was found that the accuracy dropped significantly (27%) when the anchor representation was removed and using only feature map output from the CNN to compute for the correct lane. The result matches with the Claim 2 as the anchor representation will improve the accuracy a lot.

| Dataset | Model | Backbone | Batch Size | Epoch | Accuracy % |
|---------|-------|----------|------------|-------|-----------|
| Tusimple | Line-CNN (i.e. With anchor) | resnet-18 | 8 | 20 | 87.88 |
| | Only RCNN features No Anchor at all | | | | **60.89** |

Line-CNN

Only CNN features
No Anchor at all

Figure 2: Result comparison between with and without anchor

### 4.2.2 Additional Result 2: Without Anchor pooling

The result is shown below in Figure 3. Actually, LandATT and Line-CNN are all have the part of anchor-based feature pooling. But this paper just named the Line-CNN as no anchor-based pooling. The difference is, LaneATT use all of the features alone each anchor, while the Line-CNN only use the feature at start point of each anchor. The start points are located at the left, right and bottom boders of the feature image, which means the Linc-CNN only use the border information of the feature map, while the LaneATT use the whole feature map's information.

As we can see, the LaneATT get significant higher scores than the Line-CNN. And if we look at the middle of the image, the LaneATT is much better than the Line-CNN. This proves the claim 2 that Anchor and Anchor-based pooling would increase the accuracy significantly. We can also find that the efficiency improves a lot if we only use the border information, which is very reasonable. The author got 64% F1 score. That is very close to our results 65.99% F1 score, which proves that our code implementation is correct.

### 4.2.3 Additional Result 3: Without Attention Mechanism

The result is shown below in Figure 4. We remove the attention mechanism by manually setting all the global weights to 0. The result is same as the results from the ablation study in the paper. In fact, there's very limited improvements on the detection performance. The difference on accuracy is less than 1%. However, the speed becomes significant faster if we remove the attention mechanism. Thus, combing the pros and cons, we think the attention mechanism is not as valuable as the paper claimed. So, the claim 3 from the paper may not be valid according to our results.

### 4.2.4 Additional Result 4: Cross-Entropy Loss

The result is shown below in Figure 5. It was found that the focal loss will have 1% increase in accuracy compared with the cross-entropy loss. The result is not as promising as Claim 4 stated. By exploring the potential causes of the observation, it was found that the focal loss is targeted to systems with class imbalance such as Fast R-CNN, where the feature size may vary on a large scale. However, in LaneATT, after projecting the pooled anchor features onto the feature map, the desired feature size is the same for all classes, therefore using focal loss does not show stunning improvements compared with cross-entropy loss. The additional hyperparameter introduced by focal loss may also slightly influence the computation efficiency. Therefore, Claim 4 from the paper is rejected as either focal loss or cross-entropy loss can work perfectly in LaneATT.

| Dataset | Model | Backbone | Batch Size | Epoch | F1 % | Precision % | Recall % | Speed items/s |
|---------|-------|----------|-----------|-------|------|-------------|----------|---------------|
| Culane | LaneATT (with Anchor-base pooling) (Using the whole feature map) | resnet-34 | 8 | 15 | 76.67 | 83.02 | 71.22 | 13.5 |
| | Line-CNN (Using only the feature at start points) | | | | **65.99** | **71.86** | **61** | **15.79** |

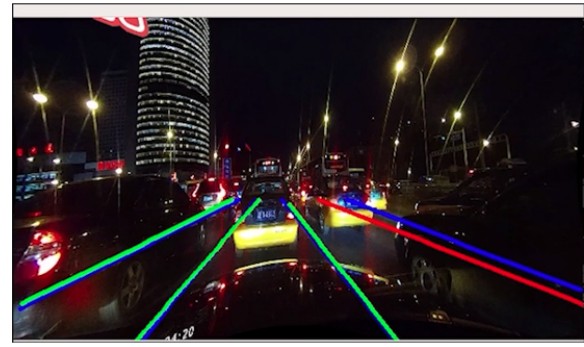
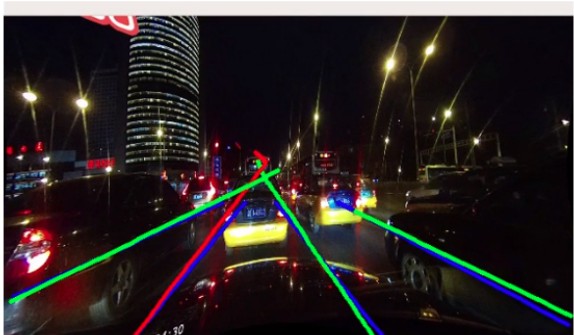

LaneATT
with Anchor-base pooling
Using the whole feature map

Line-CNN
Using only the feature at start points

Figure 3: Result comparison between with and without anchor-based feature pooling

| Dataset | Model | Backbone | Batch Size | Epoch | F1 % | Precision % | Recall % | Speed items/s |
|---------|-------|----------|-----------|-------|------|-------------|----------|---------------|
| Culane | Original model | resnet-34 | 8 | 15 | 76.67 | 83.02 | 71.22 | 13.5 |
| | Model without attention mechanism | | | | **75.75** | **83.05** | **69.62** | **17.72** |

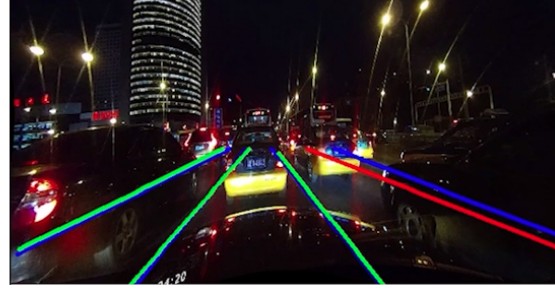
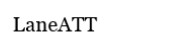
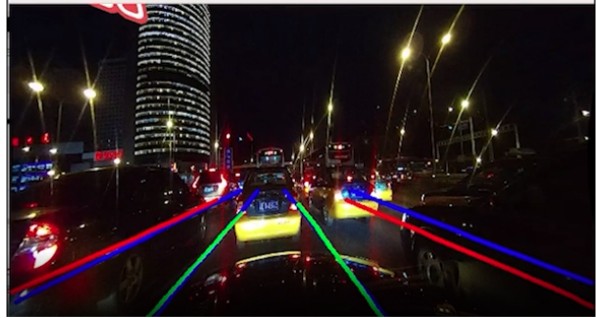

LaneATT

LaneATT
without Attention Mechanism

Figure 4: Result comparison between with and without attention mechanism

### 4.2.5   Additional Result 5: Different Backbone

The result is shown below in Figure 6. We replace the resnet with VGG-16, which is pretrained on ImageNet. We found that the model could also easily got great scores and performance well in complex city scenrios. Moreover, the VGG backbone we use is only 16 layers. It's more light, which means the anchor-based method could be possible to use lighter backbones. More robustness helps to more efficiency and feasibility.

| Dataset | Source | Backbone | Batch Size | Epoch | Accuracy % |
|---------|--------|----------|-----------|-------|-----------|
| Tusimple | LaneATT Otiginal model | resnet-18 | 8 | 20 | 93.9 |
| | LaneATT Replace focal loss with corss entropy | | | | **92.94** |

Figure 5: Result comparison between focal loss and cross-entropy loss

| Dataset | Model | Backbone | Batch Size | Epoch | F1 % | Precision % | Recall % | Speed items/s |
|---------|-------|----------|-----------|-------|------|------------|----------|---------------|
| Culane | Original model | resnet-34 | 8 | 15 | 76.67 | 83.02 | 71.22 | 13.5 |
| | Self-trained model | VGG-16 pre-trained with batch normalization | 8 | 15 | **71.9** | **81.04** | **64.62** | **17.72** |

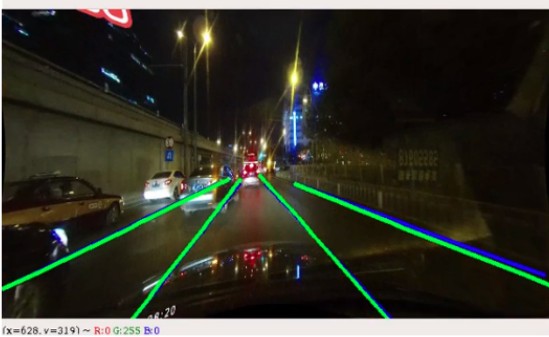

VGG-16

Figure 6: Result comparison between ResNet and VGG

## 5 Discussion

The result of shows that the the same result of the code can be obtained according to the paper's configuration. The experiments also shows that the anchor-based feature method is significantly beneficial on accuracy. While the attention mechanism can only bring slight improvement in accuracy with less real-time efficiency. The focal loss does not have significantly better performance while the Cross-Entropy loss can also work perfectly. Therefore Claim 1 and 2 are confirmed while Claim 3 and 4 can be rejected based on application (seeking higher accuracy or efficiency).

### 5.1 What was easy

The code is already open sourced and the same result can be obtained according to the paper's configuration on TuSimple and Culane datasets. The setup is quite simple following the instruction of the open source code.

### 5.2 What was difficult

The annotations for the testing dataset of LLAMAS can not be obained, therefore the experiment on this dataset can not be done.

While conducting further experiment of anchor-based feature pooling, the ablation experiment is nearly the same as Line-CNN, however it is not open sourced. Therefore, modification of LaneATT code is required for this experiment.

### 5.3 Communication with original authors

None.

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
