# OpenReview forum: "Reproducibility Report of LaneATT"
_ML_Reproducibility_Challenge/2021/Fall — Reject_

### Official Review · Reviewer_EJym · 2022-03-07
**Review for submission #3**

**Rating:** 6
**Confidence:** 3

**Review:**

The report reproduces the work "Keep your Eyes on the Lane: Real-time Attention-guided Lane Detection".

What was good:
1. The authors did a reasonable job in reproducing the original work. In particular, the authors tested the original hyperparameters plus some additional ones such as image size and number of anchors, which demonstrate non-negligible impacts on the model accuracy.
2. The authors also tested new variations/ablations of the original method, which provides additional information for judging the validity of several claims made in the original work.

What could be better:
1. For the challenge encountered, it could be helpful to reach out to the original authors for advice.
2. It would be helpful if it could be made more clear like in a summarized paragraph the part that was implemented in the report compared to the original repository.
3. The paper contains author names, which makes it non-anonymous.

---

### Official Review · Reviewer_LMbc · 2022-04-06
**An extensive report with easily achievable improvements.**

**Rating:** 7
**Confidence:** 3

**Review:**

Reproducibility Summary : This report includes a summary which contains its major findings.
Scope of Reproducibility : The report shares its scope of reproducibility and adheres to it.
Code : The report uses the same code as the author. They provide the link for the same.
Communication with original authors : None
Ablation study : The authors provide four different well motivated results.
Discussion on results : The report mentions about the difficulty in performing ablations and missing dataset.

There are few grammatical issues :
1) " detection method, . It allows"
2) " all state-of-the-artmethods"
and some more which need correction.

---

### Meta-Review · Area_Chair_Jfs2 · 2022-04-09

**Recommendation:** Reject
**Confidence:** 4

**Metareview:**

A good reproducibility study where the authors tested the approach not only on the hyperparameters provided by the original paper but also on different hyperparameters validating the robustness of the method.  However, the authors revealed their identities in the submission which violates the double-blind policy. Our policy is to desk reject papers that do not conform to submission policies, which includes violation of double-blind. However, we are considering this submission as a very special case as the authors only disclose their names but not their affiliations. Hence we are rejecting the paper and not desk-rejecting it.

---

### Decision · Program_Chairs · 2022-04-09

Reject